# Successful Treatment of an Acinar Pancreatic Carcinoma in an Inland Bearded Dragon (*Pogona vitticeps*): A Case Report

**DOI:** 10.3390/ani14131976

**Published:** 2024-07-04

**Authors:** Johannes Hetterich, Marion Hewicker-Trautwein, Wencke Reineking, Lisa Allnoch, Michael Pees

**Affiliations:** 1Department of Small Mammal, Reptile and Avian Medicine and Surgery, University of Veterinary Medicine Hannover, Foundation, 30559 Hannover, Germany; 2Department of Pathology, University of Veterinary Medicine Hannover, Foundation, 30559 Hannover, Germany

**Keywords:** lizard, neoplasia, pancreatic disease, ultrasonography, histopathology, coeliotomy

## Abstract

**Simple Summary:**

This report describes the diagnostic evaluation and therapy of a bearded dragon with an irregular mass within its body cavity. Initially, the eight-year-old lizard was presented with unspecific clinical symptoms (reduced overall condition and reduced forage intake). A round-shaped mass was determined by physical examination. The irregular structure subsequently was visualized by ultrasonography. After two days of stabilization therapy in the clinic, the mass was removed surgically under general anesthesia. The histological examination of the irregular tissue revealed a pancreatic tumor. Neoplasms of the pancreas are rare in lizards, and successful treatment and long-term survival are not commonly reported. The lizard recovered slowly but gradually after the surgical procedure and regained regular forage intake and behavior within three weeks. Long-term survival was confirmed by follow-ups within two years after the surgery.

**Abstract:**

An adult, 362 g, male, intact inland bearded dragon (*Pogona vitticeps*) was admitted to a veterinary clinic due to a temporary cloacal prolapse and a two-week history of reduced overall condition and forage intake. Physical examination revealed an approximately 2 × 1 cm round-shaped, rigid intracoelomic tissue mass. Multiple sand deposits were present on the cloacal mucous membranes, though no signs of cloacal prolapse were present. The lizard was otherwise responsive but showed reduced body tension and movement behavior. Initial fecal examination revealed a high-grade oxyuriasis. A 2 × 1.5 cm sized intracoelomic, well-vascularized, round-shaped mass was subsequently visualized by ultrasonography. After a two-day stabilization therapy, the intracoelomic mass was removed by performing a standard ventral coeliotomy under general anesthesia. Histopathological examination of the excised mass revealed an acinar pancreatic adenocarcinoma with infiltration of the peritumorous connective soft tissue. The lizard remained at the clinic for a further seven days. Its postsurgical condition improved slowly. However, the lizard started regular forage intake 10 days after surgery, and general behavior enhanced constantly within the following three weeks. The animal was presented for a follow-up six weeks after surgery, showing bright and alert behavior with no signs of disease or illness. The lizard was re-examined 20 months after the initial presentation due to a reduced overall condition and reduced food intake. Blood chemistry evaluation revealed markedly decreased protein parameters, and moderate ascites was identified ultrasonographically. A distinct association with the preceding neoplastic disease could not be made, and the lizard returned to its regular condition under supportive therapy within three weeks. To the authors’ knowledge, this is the first report of successful treatment of a pancreatic carcinoma in a bearded dragon.

## 1. Introduction

Diagnosis of pancreatic disease in reptiles is challenging, as clinical signs are often nonspecific and indicative signs, like pronounced changes in blood parameters, are lacking compared to mammalian medicine [1]. Accurate diagnoses of pancreatic diseases in reptiles are oftentimes based on postmortem examinations.

Pancreatic neoplasia reports in reptiles are rare [2]. For lizards, multihormonal islet cell carcinomas have been described for Komodo dragons (*Varanus komodoensis*) [3]. A pancreatic glucagonoma was found in a rhinoceros iguana (*Cyclura c. figgensi*) [4]. A retrospective study of neoplasms of captive lizards in the United Kingdom evaluated 158 tumors, revealing a prevalence of pancreatic neoplasia in 1.3% of all tumor diseases [5]. Survey-based data on long-term outcome of pancreatic neoplastic disease in lizards are not available. Disease is usually found to be in an advanced stage by the time of presentation [1]. Clinical symptoms like lethargy, anorexia and weakness are pronounced but nonspecific [1]. Treatment attempts should include nutritional support, fluid therapy, and analgesia [1,2]. However, most animals are euthanized due to severe clinical conditions and poor prognosis [1]. 

## 2. Clinical Case

### 2.1. History and Clinical Examination

An eight-year-old, 362 g, male inland bearded dragon (*Pogona vitticeps*) raised under human care was presented to the veterinary clinic due to a temporary cloacal prolapse. Also, the animal owner reported a three-week history of progressing lethargy and reduced forage intake. Husbandry conditions included a 180 × 60 × 60 cm glass terrarium with a temperature gradient of 24–29 °C (75–85 °F) and a basking area of 35 °C (95 °F). The animal owner provided an ultraviolet light source for 10 h/day and maintained a 40–50% humidity within the terrarium. The daily diet included various herbs, salads, and vegetables. Insects (crickets, cockroaches) were given twice a week and were constantly dusted with supplemental calcium powder. The animal owner did not report any pre-existing disease or medical problem in the lizard.

Clinical examination revealed a good body condition. However, the lizard showed reduced body tension and movement behavior. An approximately 2 × 1 cm round-shaped, rigid intracoelomic tissue mass in the middle third of the medial right coelom was identified by coelomic palpation. Minor sand deposits were stuck at cloacal mucous membranes, though no signs of cloacal prolapse were present. Based on the general examination, no other abnormalities than those mentioned above were present. Upon case history and general examination, neoplastic disease of different coelomic organs (intestinals, spleen, liver, gallbladder, pancreas) and an intracoelomic abscess were the main differential diagnoses. 

### 2.2. Diagnostic Procedures

Initial diagnostic methods included parasitological fecal examination, radiography, and ultrasonography. The animal owner declined a blood examination due to financial restrictions. Microscopic fecal examination at 100× magnification revealed a high infestation of oxyurids (>20 eggs per field of view) and a low-grade flagellate infestation. Under manual restraint, dorsoventral and horizontal beam lateral radiographic projections were performed (Figure 1; digital X-ray; detector system: Fujifilm Console Advance DR-ID 300 CL, Fujifilm Europe GmbH, Düsseldorf, Germany; tube system: Gierth X-ray International GmbH, Riesa, Germany; film focus distance FFD 60 cm, 50 kV, 5 mAs). The lateral radiograph was insufficient for any evaluation due to a technical issue. On the dorsoventral view, the medial right coelom was filled with soft-tissue material and hypodense areas (moderately filled gastrointestinal tract and gastrointestinal gas). Mineral opacities were visible in the left coelom, indicating deposits of foreign material (sand or similar anorganic substrate) in the large intestine. Both lung fields were visible on the dorsoventral radiograph, though superimpositions with the gastrointestinal tract did not allow a clear evaluation of the entire lung field size. No distinct intracoelomic lesion was determinable upon radiography. Ultrasonographic examination under manual restraint and in dorsal recumbency (Micro curved array transducer, 5–9 MHz; Vivid 7 Dimension; GE Healthcare GmbH, Solingen, Germany) with transverse and longitudinal views from the ventral coupling site showed an approximately 2 × 1.5 cm round-shaped, intracoelomic mass located in the middle third of the medial right coelom. The structure was surrounded by intestinal and fat body tissue (Figure 2). However, the mass showed good vascularization, and demarcation to surrounding tissue was clearly feasible (Figure 3). No other abnormalities of coelomic organs were visualized during the ultrasound examination. 

Based on the ultrasonographic findings and the reduced clinical condition, the presumptive diagnosis for the intracoelomic mass was a neoplastic disease involving the gastrointestinal tract. Distinct organ affiliation could not be interpreted at this stage. Further diagnostic imaging, including a coelioscopy (and, if possible, a biopsy of the affected tissue), might have been reasonable. However, the animal owner declined any further diagnostic testing. An explorative coeliotomy as a treatment attempt was conducted. The animal preoperatively was treated with fluid administration (20 mL/kg SC SID; Sterofundin, ISO 1/1 E, B. Braun Melsungen AG, Melsungen, Germany), analgetic therapy (0.3 mg/kg SC SID; Meloxicam; Metacam 2 mg/mL, Boehringer Ingelheim GmbH, Ingelheim, Germany), and housing in the species’ preferred optimal temperature zone (POTZ) over a period of two days.

General anesthesia was induced with hydromorphone (1 mg/kg IM; Hydromorphon 2 mg/mL, hameln pharma GmbH, Hameln, Germany), ketamine (10 mg/kg IV; Ketamin 100 mg/mL, CP-Pharma Handelsgesellschaft mbH, Burgdorf, Germany) and medetomidine (0.1 mg/kg IV; Cepetor 1 mg/mL, CP-Pharma Handelsgesellschaft mbH). The bearded dragon was intubated with a modified 14-gauge IV catheter. Anesthesia was maintained using inhalant isoflurane (1–3%) and oxygen (0.8 L/min) via intermittent positive-pressure ventilation. Anesthesia monitoring included monitoring the heart and respiratory rate as well as cloacal temperature. Thermal support was provided throughout the surgical procedure. The lizard was placed in dorsal recumbency and, after a standardized aseptical preparation using an iodine solution (Jodosept^®^, Vetoquinol GmbH, Ismaning, Germany) and chlorhexidine, a coeliotomy using a ventral paramedian approach was performed. After a skin incision with a scalpel blade (#11), the ventral abdominal muscles and the peritoneal membrane were incised stepwise using Metzenbaum scissors. A prominent, round-shaped, well-vascularized mass in the medial right mid-coelom was identified. The tan-colored bulging tissue with an uneven but smooth surface was closely related to the small intestines and even more closely adjacent to pancreatic tissue. However, no capsule-like demarcations were visibly connected to further organs and the mass was resected completely. An extensive hematoma was present in between the pancreatic tissue, small intestines, and the irregular mass lesion. It was removed partly using Metzenbaum scissors, as its inseparable connection with the surrounding small intestine and pancreatic tissue did not allow for a complete extirpation. No resection or incision of regular small intestine and pancreatic tissue was performed. The surgical procedure was proceeded by a thorough coelomic inspection and lavage with sterile warm fluids (Sterofundine ISO 1/1 E, 15 mL; B. Braun AG, Melsungen, Germany). Conclusively, stepwise standard closure of the coelom with a continuous over-and-over one-layer suture for the peritoneal membrane and ventral abdominal muscles (Polyglycolide, coated, absorbable), as well as simple interrupted skin sutures (Polyglycolide, coated, absorbable), was performed.

### 2.3. Postoperative Care and Follow-Up

The lizard recovered slowly from anesthesia. Despite administering a reversal agent of atipamezole (0.1 mg/kg IM; atipamezole 5 mg/mL, CP-Pharma Handelsgesellschaft mbH, Burgdorf, Germany) and fluids (20 mL/kg SC BID, Sterofundin, ISO 1/1 E) postoperatively, the lizard remained unresponsive, with markedly decreased reflexes and no signs of independent movement behavior for the first 18 h postoperatively. However, fluids (20 mL/kg SC BID; Sterofundin, ISO 1/1 E) and analgetic therapy (Meloxicam 0.3 mg/kg SC SID; Tramadol, 10 mg/kg every 48 h IM; Tramadolhydrochloride 50 mg/mL, Dechra Veterinary Products GmbH, Aulendorf, Germany) were proceeded and an antibiotic treatment (10 mg/kg SID SC; Enrofloxacin, Baytril 25 mg/mL, Elanco GmbH, Cuxhaven, Germany) was started. The patient’s behavior improved continuously on day one post surgical procedure. 

On the second day after surgery, the animal’s body tension and reflexes returned to their full extent. The animal also regained independent movement behaviour. Postsurgical therapy for three days included the fluid, analgetic and antibiotic treatment mentioned above as well as supportive care with sucralfate (50 mg/kg SID PO; Sucrabest^®^, Combustin GmbH, Hailtingen, Germany), forced feeding (15 mL/kg SID PO; Oxbow Critical Care Herbivore, Oxbow Pet Products, Murdock, NE, USA) and UV-radiation therapy for 15 min daily. The lizard was released from the clinic three days after surgery. The owner continued oral therapy with enrofloxacin (prescribed for further ten days), meloxicam (seven days), tramadol (five days), sucralfate (seven days) and forced feeding (salad and herbs every 48 h, and crickets twice per week). Three days later, the owner presented the bearded dragon with markedly reduced movement behavior and body tension and anorexia. Blood sampling and a follow-up ultrasonography were recommended to the animal owner but rejected due to financial restrictions. The patient was treated in the clinic for a period of another three days, continuing the therapy mentioned, which included enrofloxacin (10 mg/kg SID SC), Meloxicam (0.3 mg/kg SC SID), tramadol (10 mg/kg every 48 h IM), fluids (20 mL/kg SC BID; Sterofundin, ISO 1/1 E), sucralfate (50 mg/kg SID PO) and forced feeding (daily). Its general condition improved, and one week after release from the clinic the owner reported slow but constant progress in the animal’s condition. The lizard regained regular food intake after 16 days. Its overall condition was evaluated as unremarkable in a follow-up six weeks after surgery. The healing of the incision site on the ventral coelom was unremarkable, and all skin sutures were removed. 

The lizard was re-examined 20 months after the initial presentation due to a reduced overall condition and appetite. Clinically, it showed nonspecific symptoms, including a reduced body tension and moderately weakened overall condition. Based on the general examination, no other abnormalities than those mentioned above were present. Ultrasonography of the coelom under manual restraint in dorsal recumbency identified moderate ascites, but no other distinct findings relating to any of the coelomic organs were found. Also, the coelomic area of the previous surgery revealed no abnormalities. Blood collection from the ventral tail vein was performed using a 22G cannula (0.5 mL, lithium-heparine). Blood chemistry evaluation revealed markedly decreased protein parameters (total protein 1.95 g/dL (3.0–8.1 g/dL), albumin 1.02 g/dL (1.2–4.0 g/dL), references according to Exotic Animal Formulary, J. Carpenter, fifth edition, 2018). All other blood parameters were evaluated to be within the species-specific references. Differential diagnoses included various liver and gastrointestinal diseases. Any further plausible diagnostic procedures (complete blood cell count, ascites sampling, coelioscopy, magnetic resonance imaging) including stationary therapy were declined by the animal owner. A supportive therapy for the lizard included furosemide (5 mg/kg SID PO; furosemide 50 mg/mL, cp-pharma, Handelsgesellschaft mbH, Burgdorf, Germany), a liver-supporting product (50 mg/kg SID PO; Legaphyton^®^, Vetoquinol S.A., Lure, France), and forced feeding of a high-protein diet (raising the number of cockroaches and crickets from 2–3 up to 7–8 weekly). The lizard’s condition improved, and the animal regained regular food intake three weeks later. The animal owner neither presented the lizard for the follow-up appointment four weeks later nor any later follow-up. 

### 2.4. Histological, Immunohistochemical and Electron Microscopical Evaluation

The surgically removed mass was fixed in 10% neutral buffered formalin, routinely embedded in paraffin, and 2–3 µm sections were stained with hematoxylin and eosin (H&E) for histologic examination. 

Histological examination of the completely excised tissue revealed a well-differentiated moderately cell-dense demarcated multinodular neoplastic mass with an acinar growth pattern that extended multifocally into the organ capsule (Figure 4A,B). The mass replaced normal pancreatic architecture. Upon multiple cross sections, no endocrine islets were detectable. The multifocally dilated acinar structures consisted of cuboidal to columnar tumor cells. Throughout the tumor, a strong eosinophilic, slightly granular staining of the apical cytoplasm was seen, indicating the presence of cytoplasmic zymogen granules (white arrows, Figure 4C). The medium-sized or large nuclei of the tumor cells were located in the periphery of the cells and had prominent nucleoli (black arrows, Figure 4C). Sometimes tumor cells with vacuolated cytoplasm and vesicular appearing nuclei were found (black arrowheads, Figure 4C). There was moderate anisocytosis and anisokaryosis with nucleolar pleomorphism. Mitotic figures were not seen. Between the tumor cells, a scant amount of fibrovascular stromal tissue was present. The mass was surrounded by a rim of connective tissue (interpreted as organ capsule) which had multiple hyperemic blood vessels, a mild focal hemorrhage, and multifocal infiltration by tumor cells (Figure 4D). Based on the histomorphology, an acinar pancreatic carcinoma was suspected.

To confirm the diagnosis, additional testings have been applied to the samples. 

Periodic acid-Schiff (PAS) reaction and PAS reaction with diastase digestion were used for demonstration of zymogen granules. Immunohistochemistry (IHC) on paraffin sections was carried out using the avidin−biotin complex (ABC) technique and primary mouse monoclonal antibodies to pan-cytokeratin (panCK, clone AE1/AE3: 1:500), vimentin (clone V9: 1:100), and neuroendocrine markers, i.e., synaptophysin (clone DAK-SYNAP; 1:500), and neuron-specific enolase (NSE, clone BBS/NC/VI-H14: 1:100). In addition, a rabbit polyclonal antibody against bovine chromogranin A (bovine pituitary secretory protein chromogranin, SP1 CrgA: 1:2000) was applied. All antibodies were purchased from Dako Deutschland GmbH, Hamburg, Germany (now Agilent Technologies GmbH, Waldbronn, Germany). As secondary antibodies, biotinylated goat anti-mouse and goat anti-rabbit antibodies (each 1:200) were used. Signal enhancement was achieved with the avidin−biotin−peroxidase complex kit (VECTASTAIN^®^ Elite^®^ ABC-Kit, Vector Laboratories, Inc., Burlingame, CA, USA). Peroxidase activity was detected using 0.1% H_2_O_2_ with 3,3′-diaminobenzidine solution as the chromogen. Sections were counterstained with Mayer’s hematoxylin. All immunohistochemical reactions included positive control archival tissue of bearded dragon intestine and pancreas, as well as appropriate canine tissues. As negative controls, respective dilutions of immunoglobulin isotypes or rabbit serum replaced the primary antibodies (Appendix A).

All H&E and IHC slides were digitized using the Olympus VS200 digital slide scanner (Olympus Deutschland GmbH, Hamburg, Germany). Representative images of slides were exported with the OlyVIA software (Olympus Deutschland GmbH, Hamburg, Germany; www.olympus-lifescience.com/de, accessed on 2 March 2024).

Transmission electron microscopy was performed by re-embedding H&E-stained paraffin sections in epoxy resin using the pop-off technique described by Lehmbecker et al. [6]. Slides were evaluated with an EM906-transmission electron microscope (Carl Zeiss CMP GmbH, Göttingen, Germany). Pictures were recorded on a CCD-digital camera (TRS Tröndle Restlichverstärkersysteme, Moorenweis, Germany).

The PAS reaction showed that PAS-positive zymogen granules were present in the cytoplasm of tumor cells and that their numbers varied from numerous to frequent between different areas of the tumor. Also, the number of cells with diastase-resistant cytoplasmic zymogen granules varied from frequent to uncommon between different areas of the tumor. Immunohistochemically, tumor cells showed a positive diffuse centroacinar immunolabeling for panCK. (Figure 4E). In addition, multifocally single to few acinar tumor cells showed a homogenous cytoplasmic staining. No labeling was observed with all other applied markers (vimentin, NSE, CrgA, synaptophysin). Immunolabeling for pan cytokeratin resembled the pancreatic pattern observed in the control tissue of another bearded dragon (insert, Figure 4E). Altogether, the findings of the immunohistochemical examination mentioned above confirmed the diagnosis of an acinar pancreatic carcinoma.

Transmission electron microscopical examination revealed the presence of zymogen granules in the apical cytoplasm of tumor cells. Zymogen granules were distinct, round, and had a homogenous and sometimes fine granular appearance. They showed a slight to medium electron density, and their number varied between different tumor cells (ZG, Figure 4F). In addition, high amounts of rough endoplasmic reticulum are present in the tumor cells (rER, Figure 4F). 

## 3. Discussion

Diseases of the reptile pancreas are not commonly reported. However, several differential diagnoses should be considered relating to the case reports in the literature [1]. Herpes viral inclusion bodies, as well as arenavirus and ferlavirus related inclusion bodies, have been found to affect the pancreas in snakes [7,8]. Also, differential diagnoses of pancreatic neoplasia include pancreatitis, which has also been described for reptile species [2]. Acute necrotizing pancreatitis, as in mammals, can be a result of the escape and activation of pancreatic enzymes, leading to severe inflammation processes [1]. Depending on the location and severity of the inflammation, both exocrine and endocrine portions of the pancreas may be involved. Also, pancreatitis can be described as being associated with endoparasitic infection [2]. Migratory nematodes and trematodes might cause damage to pancreatic tissue. They might also trigger secondary bacterial inflammation processes. Validated blood chemistry values might be highly beneficial as elevated enzyme parameters could serve as pancreatitis markers. However, validation of these enzymes has not yet been established for reptiles. Therefore, diagnostic imaging methods, mainly ultrasonography and endoscopy (coelioscopy), represent important diagnostic tools for further verification of pancreatitis and other pancreatic disorders.

In the present case, ultrasonography was used for diagnostic clarification. Ultrasound examination represents a valuable tool for the assessment of bearded dragons and the visualization of most, but not all, coelomic structures [9]. However, a clear association of the intracoelomic mass to a specific organ could not be determined with the imaging techniques used. Further imaging techniques have not been established to reliably identify the pancreas in bearded dragons. A coelioscopy, including taking a biopsy of the affected tissue, might have been a reasonable diagnostic tool for further diagnostic workup. Nevertheless, a subsequent explorative coeliotomy was preferred by the authors due to the poor overall condition of the lizard.

The pancreatic gross anatomy of lizards is described as being trilobed, with each lobe extending toward the gallbladder, duodenum, and spleen, respectively [10,11]. Intraoperatively, the mass lesion’s close anatomical position adjacent to pancreatic tissue and the small intestines made different tentative diagnoses feasible. Pancreatitis appeared unlikely due to the physiological gross anatomy of the visible pancreatic tissue. Small intestine neoplasia was a potential diagnosis due to the lesion’s anatomical localization. However, intact intestinal passage and unremarkable integrity of the intestines did not support this option. A tentative diagnosis of pancreatic neoplastic disease was made due to the anatomical localization and the heterogeneous, well-vascularized texture of the tissue mass most closely related to the pancreatic tissue. Grossly, the surgically removed pancreatic carcinoma of this bearded dragon appeared as a focal, solitary, round-shaped, tan-colored, bulging, nodular mass with an uneven but smooth surface. These macroscopical features resemble those described for acinar pancreatic carcinoma in humans, dogs, and cats [12,13,14]. Based on the histopathological findings, demonstration of zymogen granules by PAS and TEM and the immunohistochemical positivity of tumor cells for panCK obtained on the surgically excised mass, a well-differentiated acinar pancreatic carcinoma was diagnosed. According to the literature, cells of acinar cell carcinoma of the pancreas contain cytokeratins 8 and 18, and these two cytokeratins react with broad spectrum cytokeratin antibodies such as AE1/AE3, which has been demonstrated in human, canine, and feline acinar cell pancreatic carcinomas [12,14,15,16,17,18].

In addition, the lack of immunolabeling for neuroendocrine markers favors the diagnosis of an exocrine pancreatic tumor. The infiltration of tumor cells through the capsule in the present case is indicative of a malignant tumor, as is reported in domestic mammals [16]. Due to the necessity of evaluating capsular and/or vascular invasion, the total excision of detected potential endocrine tumors should be preferred. Due to the malignant characteristics of the pancreatic tumor, the risk of metastatic spread was conclusive. An ideal follow-up management of the animal in this report would have comprised regular follow-up examinations, including ultrasound, radiographic and computed tomography examinations for screening any recurring neoplastic (and metastatic) tumor emergences [16]. 

Retrospective studies have shown that exocrine pancreatic carcinomas are rare in both older dogs and cats [13,14]. In contrast to humans, the majority of canine and feline pancreatic carcinomas have an acinar growth pattern, while ductal carcinomas occur less frequently [13,14]. As described for acinar pancreatic carcinomas in humans and cats [12,14], and in a report about a pancreatic clear acinar cell carcinoma in one dog [18], the acinar tumor cells of the pancreatic carcinoma in the bearded dragon examined in this study expressed panCK. Ultrastructurally, zymogen granules were demonstrated in the tumor cells of the neoplastic tissue of this bearded dragon, as already reported for humans and one dog with acinar pancreatic carcinoma [12,17]. Besides histology, positive PAS reaction and positivity for panCK, the presence of intracytoplasmic zymogen granules in tumor cells is described as being one of the most important diagnostic features of acinar pancreatic carcinomas in humans [18] and was also described in one dog [17]. 

Several reports describe the occurrence of neoplasms in the alimentary tract of bearded dragons, including different types of gastric and intestinal carcinomas, most frequently being neuroendocrine carcinomas [19,20,21]. Furthermore, reports on one case each of a gall bladder carcinoma [22] and a round cell tumor of liver and spleen exist [23]. In the retrospective study by Kubiak et al. [5], only three of 43 neoplasms affecting the alimentary tract of lizards were of pancreatic origin. In one of these three cases, a round cell sarcoma was present in the pancreas and intestines of a bearded dragon. In two animals, belonging to a different species of lizard, an undifferentiated carcinoma and an islet cell carcinoma of the pancreas were found, respectively. However, in none of the aforementioned reports described a pancreatic carcinoma. Database searches (Web of Science, Scopus, CABI, PubMed and Google Scholar) for ‘neoplastic disease in reptiles’, ‘tumor and metastasis’, ‘pancreatic carcinoma in bearded dragons’ covering the years 1924–2023 were negative for a report of a pancreatic carcinoma in a bearded dragon. 

The lizard’s clinical condition 20 months after the initial presentation with moderate ascites and marked hypoproteinemia might be related to re-emerging pancreatic and/or further gastrointestinal disorders. However, etiologies for ascites and hypoproteinemia are diverse, including liver, kidney, or cardiopulmonary disease [1]. Due to the patient’s clinical improvement under supportive therapy and the financial impairments of the animal owner, no further diagnostic methods were conducted to evaluate possible etiologies.

## 4. Conclusions

Pancreatic neoplasms are rarely diagnosed in bearded dragons. Establishing a profound diagnosis is challenging. However, pancreatic neoplasia should be considered as a differential diagnosis for intracoelomic mass lesions. High-risk treatment attempts can be rewarding if they utilize surgical extirpation of neoplastic tissue and thorough perioperative care.

## Figures and Tables

**Figure 1 animals-14-01976-f001:**
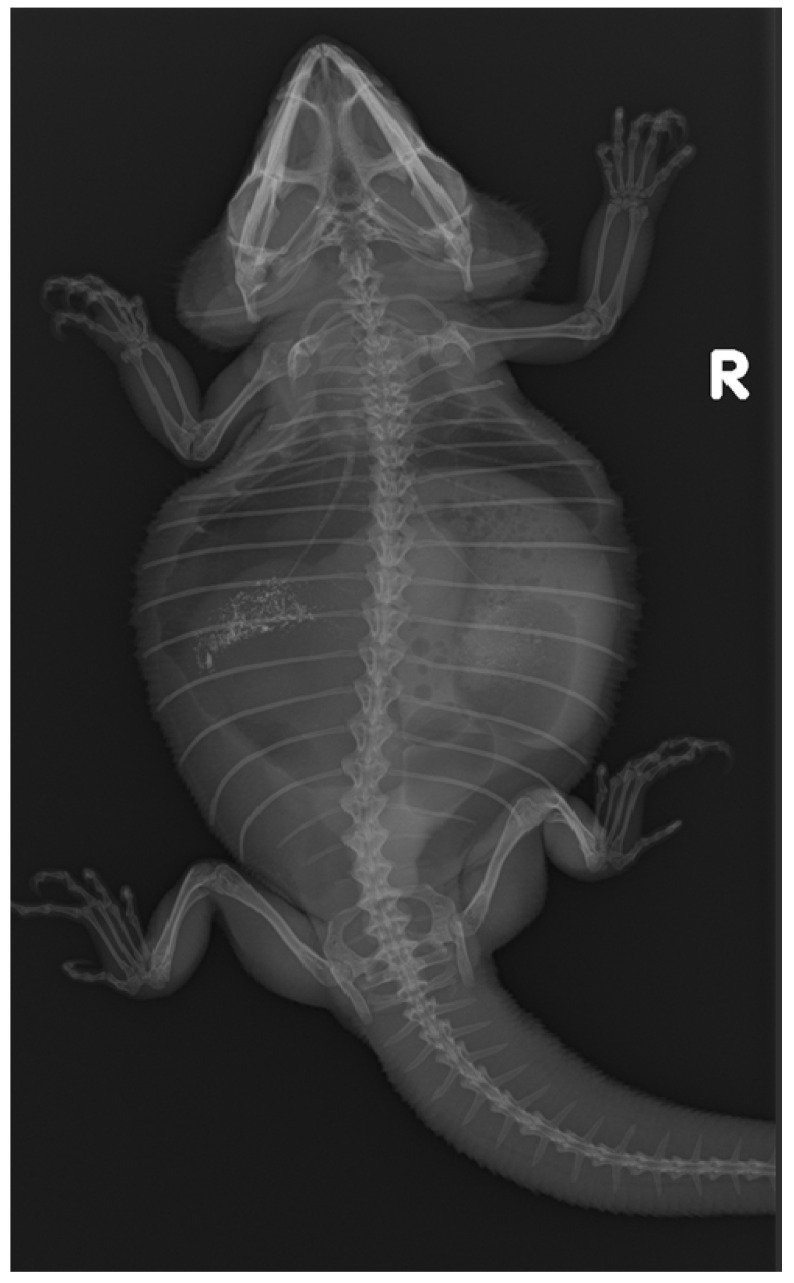
Dorsoventral radiograph of an eight-year-old bearded dragon (*Pogona vitticeps*). No indicative coelomic alterations are visible.

**Figure 2 animals-14-01976-f002:**
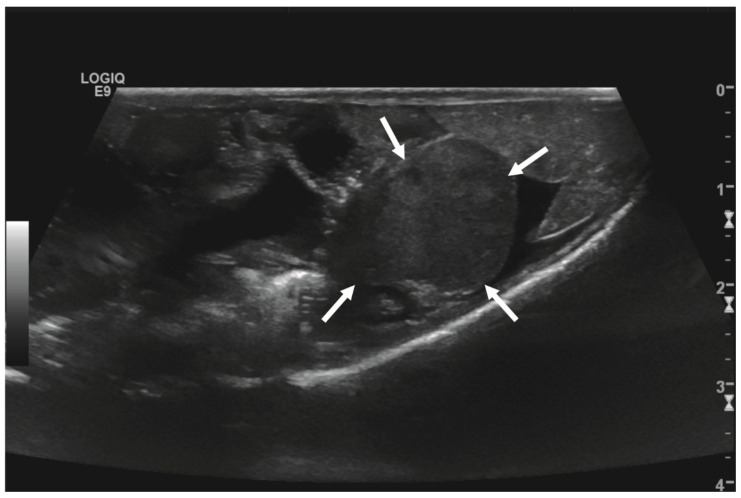
Transverse ultrasonographic image of the medial right coelom in a bearded dragon acquired with a 5 to 9 MHz micro curved array transducer (penetration depth 3 cm; frequency 15 MHz) showing an approximately 2 × 1.5 cm round-shaped, irregular intracoelomic mass (arrows). Additionally, the gastrointestinal tract (medial, here: to the left of the mass) and fat body (lateral, here: to the right of the mass) are visible and closely related to the mass.

**Figure 3 animals-14-01976-f003:**
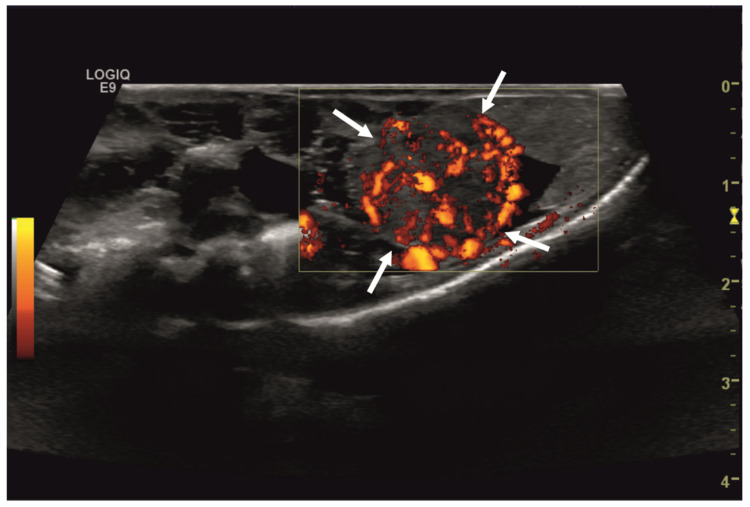
Transverse ultrasonographic image of the medial right coelom in a bearded dragon acquired with a 5 to 9 MHz micro curved array transducer and Color Doppler evaluation (penetration depth 2 cm; frequency 15 MHz). Note the distinct vascularization of the round-shaped intracoelomic mass (arrows). Additionally, gastrointestinal tract (medial, here: to the left of the mass) and fat body (lateral, here: to the right of the mass) are visible and closely related to the mass.

**Figure 4 animals-14-01976-f004:**
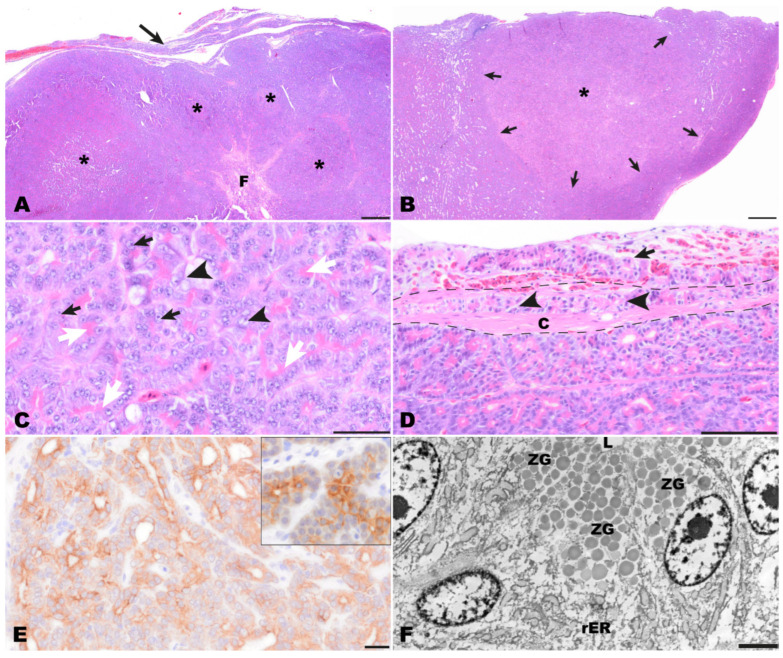
Pathomorphological findings in the surgically excised mass, bearded dragon. (**A**) Overview of the multinodular pancreatic mass (asterisks) with capsular infiltration (arrow) and central fibrosis (F). Hematoxylin-Eosin (HE). Bar = 500 µm. (**B**) Low magnification of one nodule (asterisk) highlighting that nodules are unencapsulated but well demarcated (arrows) and paler than surrounding tissue. HE. Bar = 500 µm. (**C**) Tumor cells are arranged in tubulo−acinar structures with bright pink granular material in the apical cytoplasm (white arrows). The cells exhibit prominent nucleoli (black arrows) and the partly vesicular appearance of nuclei (black arrowheads). HE. Bar = 50 µm. (**D**) Multifocally tumor cells were present within the capsule (arrowheads) as well as extracapsular adjacent to blood vessels (arrow). The dashed line indicates the thickness of the fibrous capsule (C). HE. Bar = 100 µm (**E**) A diffuse centroacinar immunolabeling of tumor for pan-cytokeratin was detected. The immunolabeling represents the reaction that can be observed in normal exocrine pancreas of bearded dragons (insert). IHC for pan-cytokeratin. Bar = 20 µm. (**F**) Ultrastructurally, epithelial tumor cells surrounding an empty lumen (L) were detected. Within the cytoplasm, large amounts of rough endoplasmic reticulum (rER) were present in the perinuclear areas. In addition, moderately electron-dense, well-demarcated granules were seen in the apical cytoplasm of the tumor cells resembling zymogen granules (ZG). Transmission electron microscopy. Bar = 2500 nm.

## Data Availability

The data presented in this study are openly available in a repository at: DOI:10.17605/OSF.IO/GDCXA.

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
