# Peer review of "Successful Treatment of an Acinar Pancreatic Carcinoma in an Inland Bearded Dragon (Pogona vitticeps): A Case Report"

_animals, 2024, doi:10.3390/ani14131976_

Round 1

Reviewer 1 Report

Comments and Suggestions for Authors

Comments to authors: This is an interesting case report, and would add to the body of literature for both surgery and neoplasia in lizards. There are numerous grammatical errors and typos present. I did not comment on most of these, as this is outside the scope of a reviewer’s responsibility. However, I did note a few examples in the beginning of the manuscript for the authors. In addition, please see the suggestions below.

Line 21: Suggest: An adult, 362-gram, male intact, inland bearded dragon…

Line 24: Multiple sand deposits were present on the cloacal mucous membranes…

Line 25: Multiple spaces present between otherwise and responsive

Line 25-26: What do the authors mean by “reduced body tension?”

Line 42: I would suggest including coeliotomy or some other surgical wording here

Line 54: Based on this wording, I would encourage the authors to include the data found on outcomes.

Line 55: Please include a reference for this statement.

Line 55-56: Please include a reference for this statement.

Line 56-57: Please include a reference for this statement.

Line 57-58: Please include a reference for this statement.

Line 61: Please include the weight of the animal here

Line 71: Please include details on heart rate, respiratory rate, mucous membrane color, capillary refill time, hydration status, perfusion parameters, etc. as well as any other abnormalities on exam (or state that there were not other abnormalities other than those described).

Line 72: As noted above, please explain what reduced body tension is.

Line 72-73: Please describe the location within the coelom as well

Line 74: What were the differentials at that time, based on history and physical examination?

Line 76: What diagnostic tests were recommended to the owner, and what was elected/declined? Having this at the beginning can ensure that the reader is not trying to anticipate what is missing.

Line 78: Please quantify the parasites – this can be done using eggs per gram or by some other method, as this will provide readers a better understanding of what was present.

Line 79: Please specify whether the lateral used a horizontal beam, and what was used for restraint (chemical, manual, etc.) and provide details here.

Line 82: Please refer to radiographic figures here. Please add in a description of the findings. Were these radiographs evaluated by a radiologist? If not, this should be performed for a full evaluation and description in the case report, as this is a large mass for such a small animal. What was used for restraint for the ultrasound? Same request as for radiographs.

Line 85: Where in the coelom was the mass? Similar to request for physical examination. Why is the measurement approximate? Was it not measured during the ultrasound?

Line 85-87: Was the ultrasound performed by a radiologist? Can the authors comment on the remainder of the examination as well please? Please also refer to the figures here. Figures 2 and 3 can likely be combined into a multi-part figure.

Line 87-88: Move this information to the beginning of the section.

Line 90: In figure one, is this radiograph labeled correctly? The stomach appears to be on the “right” side according to the marker. Please double check this, and if the marker is correct, then the radiograph needs to be flipped on the horizontal axis to make the left on the right and the right on the left (standard positioning for animals). Additionally, the radiograph is quite dark and it is challenging to evaluate. Please adjust the brightness of the image. I would encourage the authors to comment on the full stomach and the gas that is present, as well as the mineral opacities on the “left” side. Please include orthogonal views, as this is standard in veterinary medicine.

Line 94: Please include the location within the coelom, and also indicate/describe other, normal, structures for identification and orientation for the reader.

Line 103: Based on the location, what were the differentials for organ involvement?

Line 104: If the owner had financial constraints, how did they afford to perform surgery?

Line 105: Did the authors feel the animal was unstable? If so, how? The physical examination description did not suggest that. If the animal was unstable, why were SC fluids chosen and not IV/IO?

Line 109: Did the animal eat during that time? Was it assist fed? Just wondering how this connects with the patient’s instability.

Line 110: Why did the authors choose this dose of hydromorphone, as there is a PK study for its use in bearded dragons? See: Hawkins SJ, Cox S, Yaw TJ, Sladky K. Pharmacokinetics of subcutaneously administered hydromorphone in bearded dragons (Pogona vitticeps) and red-eared slider turtles (Trachemys scripta elegans). Veterinary anaesthesia and analgesia. 2019 May 1;46(3):352-9.

Line 111: Was an IV catheter attempted?

Line 118: Do the authors mean dorsal recumbency? Was the animal aseptically prepared and draped?

Line 119: Please describe the surgical procedure for the readers in more detail.

Line 120: Do the authors mean medial instead of median? Was it cranial, mid, or caudal in location? Do the authors have a gross image of the mass in surgery or of the mass after removal?

Line 124-125: Please describe the removal in more detail, as in a surgery report. There is very little on the technique and instruments used, etc. here.

Line 128: What kind of fluids?

Line 128-129: Please describe the closure in more detail, including suture used and patterns.

Line 131-132: What does this mean? Please describe this for the reader. What was attempted to get the animal to recover faster? Were any fluids administered? Were any reversal agents provided? What occurred in this 24 hour time period?

Line 135: Why did the authors administer antibiotics, and especially enrofloxacin? It is a critically important antibiotic for human medicine, and there are concerns about its use for reptiles. See: Hedley J, Whitehead ML, Munns C, Pellett S, Abou‐Zahr T, Calvo Carrasco D, Wissink‐Argilaga N. Antibiotic stewardship for reptiles. Journal of Small Animal Practice. 2021 Oct;62(10):829-39.

Line 137: Why was sucralfate initiated?

Line 138: Please include the food used, including manufacturer info, amount used, etc.

Line 139: How has the patient progressed during this time?

Line 140-142: How long were each of the medications prescribed for? What were the instructions for the sucralfate, since there were now multiple oral medications?

Line 143: What does the markedly reduced general condition statement mean? This is used a couple of times in the manuscript, but it does not describe to the reader what the authors appreciated.

Line 144: Were any fluids administered? Were feedings continued? What else was provided during this time? Was blood work offered due to patient decline?

Line 148: How was healing of the incision site? Were any sutures removed?

Line 150: Please include physical examination findings and details, as requested for the initial examination.

Line 151: What were the differentials at that time? What diagnostic tests were recommended, and elected/declined? Similar to requests for initial evaluation.

Line 151: How was the ultrasound performed – manual or chemical restraint, etc.? Same requests as for initial evaluation.

Line 152: How much blood was collected, what type of tube was used, what were the values, what reference was used for evaluation? Was a complete blood cell count offered or performed? Was a sample collected from the ascites for evaluation?

Line 153: Why was furosemide elected?

Line 154: Please describe this product in generic terms, as in what it is composed of.

Line 155-156: Please include details about the food item.

Line 157: Any additional follow up?

Line 158: This is ultimately the decision of the editor, but this section would make more sense after the surgical description and before post-op care, in my opinion.

Line 161-185: Were all of these done prior to the initial histo of the sample? These details would seem to make sense after the initial histo description, as it was needed after evaluation, correct? I would suggest moving this further down into the section.

Line 186-187: Please describe the mass prior to providing the diagnosis, as it will be based on what was found histologically.

Line 199: Can the authors please comment on margins?

Line 239-240: Please include a reference for this statement.

Line 252-253: I would include the findings regarding the pancreas from this manuscript: Bucy DS, Guzman DS, Zwingenberger AL. Ultrasonographic anatomy of bearded dragons (Pogona vitticeps). Journal of the American Veterinary Medical Association. 2015 Apr 15;246(8):868-76.

Line 255-256: This is not supported by what was described – please add details as suggested in previous sections.

Line 278-279: Please include a reference for this statement.

Line 289-290: Why did the authors underline this part of the statement?

Line 301-302: The following is in regards to the statement “to the author’s knowledge.” Please read 1. Di Girolamo N, Meursinge Reynders R. On "authors’ knowledge" and contrast-enhanced ultrasonography in rabbits. Vet Radiol Ultrasound 2019;60:371. https://doi.org/10.1111/vru.12748 . A systematic database search should be performed. I recommend you perform a comprehensive literature search using 3 databases (e.g., PubMed, CAB, Scopus [CAB gives a high yield of veterinary articles compared with PubMed]). Use the PRISMA Guideline (PRISMA. 2015; www.prisma-statement.org) and document your findings in the PRISMA flow diagram ( http://www.prisma-statement.org/PRISMAStatement/FlowDiagram ). You can then state in the manuscript, "… database searches (cite the databases you used) covering the years (e.g., 1940-2023) were negative for XXX."

Comments on the Quality of English Language

See comments in review.

Author Response

Dear reviewer,

thank you for your helpful comments on our manuscript. Please find our responses in the document attached.

Kind regards on behalf of all authors

Reviewer 2 Report

Comments and Suggestions for Authors

I enjoyed reading your manuscript, thank you. Here are some comments from me.

Is the word "round-sized" in Lines 13, 23, and 72 typo for "round-shaped"?

There seems to be an extra room btw "The lizard was otherwise" and "responsive but showed" in Line 25.

In Line 54, "pancreatic neoplasia disease" can be replaced by "pancreatic neoplastic disease" or "pancreatic neoplasia" in terms of grammar.

Is ventral recumbency in Line 118 correct? I am not familiar with surgical approach for the reptilian species but it sounded a little odd to cut open from the dorsal side to reach the abdominal mass.

“Represens” needs to be replaced by “represents” in Line 225.

Remove underline in Lines 289 and 290.

Author Response

Dear reviewer,

thank you for your helpful comments on our manuscript. Please find our responses in the document attached.

Kind regards oh behalf of all authors

Reviewer 3 Report

Comments and Suggestions for Authors

Comments on the manuscript animals-2952889 entitled “Successful treatment of an acinar pancreatic carcinoma in an inland bearded dragon (Pogona vitticeps): a case report” by Hetterich et al.

This is an interesting article describing a pancreatic neoplasm in an inland bearded dragon (Pogona vitticeps). Articles on this theme are welcome. 

Images are illustrative and clear. There are some points that need to be addressed to improve the comprehensibility of this manuscript. 

Line 78: Please clarify what authors mean with “low-grade infestation”. I´m not sure if we graduate parasite infestation.

Line 90. Figure 1. Legend: Please remove the following sentence from the legend “…presented with a three-week history of progressing lethargy and reduced forage intake.” This information is already present in the manuscript and cannot be achieved when one analyzes an X-ray.

Line 102: Please amend “…tentative diagnosis…” to “…presumptive diagnosis…”.

Line 127: Please amend “…tissue was performed.The surgical procedure…” to “…tissue was performed. The surgical procedure...”

Line 164-166: Please insert the dilution of the used antibodies.

Line 200-201: Please clarify if diastase-resistant granules were frequent or uncommon.  

Line 203: Please remove “…throughout the neoplasia…” from the sentence, as authors already reported that the CK positivity was diffuse.

Line 218: Please replace “…slightly brighter than…” to “…paler than…”

Line 226-226: Please remove “…(chromogen: 3,3’ diaminobenzidine)…” from the sentence, as authors already mentioned it in the manuscript (line 172).

Line 289-290: Please clarify the sentence “…The PAS reaction showed that numerous PAS-positive zymogen granules with lower numbers of them being resistant to diastase digestion were present…”. If few granules were resistant to diastase, probably are not zymogen granules, as zymogen granules are resistant to diastase digestion.  

Figures: Please replace the figure E, as it is not focused.

The bigger criticism I have is the use of a pan-keratin to confirm diagnosis, as most pancreatic tumors are positive to Keratin 8 and K18. Did authors try these antibodies?

In fact, the morphology is similar to “normal” pancreas, and immunohistochemistry confirmed an epithelial tumor. Electron microscopy also confirmed the morphologic diagnosis. 

Comments on the Quality of English Language

Though it is a manuscript easy to read, I think minor editing language is necessary.

Author Response

Dear reviewer,

thank you for your helpul comments on our manuscript. Please find our responses in the document attached.

Kind regards on behalf of all authors

Round 2

Reviewer 1 Report

Comments and Suggestions for Authors

Comments to authors: There continue to be multiple grammatical errors that take away from the manuscript. Again, I would encourage the authors to proofread carefully and adjust the sentence structure to make it easier for the reader to follow the progression of the case. I previously asked for more information and details regarding the thought process of the authors as they approached the case – this still isn’t present. Knowing the thought process will help the readers as they are approaching these types of cases. Please include this information where prompted, and this could also be added to the discussion section as well. Examples include – why SC and not IV fluids if the authors were concerned about the animal’s overall poor condition? Why was furosemide elected post-operatively? Why was sucralfate elected? What were the hypotheses on why the animal recovered slowly from anesthesia? These are just some examples.

Line 26: Reduced body composure is confusing wording – what are the authors attempting to convey?

Line 84: What field of view was this – 40x or 100x?

Line 85: Were both views taken with a horizontal beam? If not, I would state that it was a dorsoventral and horizontal beam lateral. I would also recommend starting the sentence with “Under manual restraint, dorsoventral and horizontal beam lateral radiographic projections were performed…”

Line 88-93: I did not see a lateral view included in the figures, and this does not read like a radiographic report. The coelom was covered? What part of the GI tract were the mineral opacities in? The DV radiograph is backwards. I didn’t see any comments about the lung fields. There is a typo before the sentence about ultrasound.

Line 93: What position was the animal in for ultrasound?

Line 96: Can the authors be more specific in the location? Cranial third, middle third, caudal third of the coelom?

Line 96-97: I do not see any reference to the figures in this area.

Line 98-99: What other coelomic organs were evaluated?

Line 118-120: I would reword this to state what was recommended and what was elected or declined. This makes it sound like further diagnostic testing was not recommended.

Line 135: What is the standard preparation? What product was used?

Line 138 and 144: Metzenbaum scissors, not “a” Metzenbaum scissor

Line 149: What do the authors mean by gradually? What kind of continuous suture pattern? Were these sutured in one layer or two layers?

Line 151: What type of interrupted skin sutures?

Line 155: What volume of fluids? What route?

Line 157: There is an extra period after postoperatively? Was partial reversal for the hydromorphone attempted?

Line 160-161: Do the authors mean the second day, since the animal didn’t move for the first 18 hours? That is most of a day.

Line 162: Body composure – see previous comments

Line 165: Why was sucralfate administered?

Line 166: What type was it – omnivore? Herbivore? Carnivore? How did the authors decide on the amount to feed? Were the nutritional requirements calculated? Was the animal offered food during this time?

Line 171: Was the owner sent home with feeding instructions? Change “he” to “the owner”

Line 172: Remove reduced general condition and leave the specifics of the movement. Change body composure.

Line 175: What dose, route, and frequency of fluids? What supportive care therapies? It may be easier to just list them by medication rather than by category so the readers are not confused and do not have to guess what was administered. Were any diagnostic tests recommended at this visit? I would state that, as well as what was thought to be going on at this point.

Line 178: What was the animal’s body weight at that time? Were diagnostics recommended at this visit due to the incomplete removal of the animal’s tumor?

Line 181: What was the animal’s body weight at that time, what was the body condition?

Line 186: What organs were evaluated during the ultrasound? What about the area of the previous surgery?

Line 188: Please include the reference interval and the reference used for interpretation. Was the rest of the blood work normal?

Line 189: The liver and GI tract were normal on ultrasound though, correct? So what were the differentials?

Line 190: Please be specific about what recommendations were made – etc. does not help the reader.

Line 191: What is in house therapy?

Line 192: Why did the authors choose to initiate furosemide?

Line 203: Above it states that the tumor was completely excised – here it says incomplete. Please change this where it is incorrect. It if was incompletely excised, what about the ultrasound that was performed? Was this area evaluated closely? It is hard to believe that there would be no tumor growth in 2 years.

Line 263: I would add a summary of the findings and what it means for the reader – basically, did all of the tests support the diagnosis?

Line 282: I would recommend mentioning something about metastatic potential of these tumors, and then relate it back to the case described. This animal’s tumor was incompletely excised – what would be the ideal medical work up for this case because of that?

Line 355: As mentioned, please remove this statement. Please also include the search terms used for the database search. I would recommend using broader search terms than just “pancreatic carcinoma in bearded dragons”

Comments on the Quality of English Language

See comments above.

Author Response

Dear Sir / Madam,

Please find the response letter attached. Thank you.

Round 3

Reviewer 1 Report

Comments and Suggestions for Authors

Thank you for the changes that the authors have made to the manuscript. 

Comments on the Quality of English Language

Additional editing needed, sufficient for copy editor to address. 

Author Response

Dear Sir/Madam,

please find the latest version of our manuscript attached.

Thank you.
